# Physiological Response of Grower African Catfish to Dietary Black Soldier Fly and Mealworm Meal

**DOI:** 10.3390/ani13060968

**Published:** 2023-03-07

**Authors:** Askale Gebremichael, Balázs Kucska, László Ardó, Janka Biró, Mária Berki, Éva Lengyel-Kónya, Rita Tömösközi-Farkas, Robert Egessa, Tamás Müller, Gergő Gyalog, Zsuzsanna J. Sándor

**Affiliations:** 1Department of Applied Fish Biology, Hungarian University of Agricultural and Life Sciences Kaposvár Campus, Guba S. u. 40, 7400 Kaposvár, Hungary; 2Department of Freshwater Fish Ecology Hungarian University of Agricultural and Life Sciences Kaposvár Campus, Guba S. u. 40, 7400 Kaposvár, Hungary; 3Research Centre of Aquaculture and Fisheries, Hungarian University of Agricultural and Life Sciences, Anna liget. u. 35, 5540 Szarvas, Hungary; 4Food Science Research Group, Institute of Food Science and Technology, Hungarian University of Agriculture and Life Sciences, Villányi út 29-43, 1118 Budapest, Hungary; 5Department of Freshwater Fish Ecology, Hungarian University of Agricultural and Life Sciences, Szent István Campus, Páter K. u. 1, 2100 Gödöllő, Hungary

**Keywords:** sustainable protein, insect meal, fatty acids, feeding, catfish

## Abstract

**Simple Summary:**

Sustainability and profitability of African catfish farming depends on a sustainable and cost-effective supply of feed. Fish meal is still used as the main dietary protein source in the practical diet of fish. However, the supply shortage and high cost are the limiting factors. Insects may represent a promising candidate for fish feeding due to their low footprint production and nutritionally relevant properties. This study revealed that replacement of fish meal with black soldier fly did not negatively affect the production performance and metabolic response of African catfish growers. On the other hand, replacement with yellow mealworm may lead to the fish’s growth reduction and health problems.

**Abstract:**

A six-week experiment was carried out to test the effects of total (100%) and partial (50%) replacement of fish meal in the diet of African catfish growers with black soldier fly (B) meal, yellow mealworm (M) meal, and a 1:1 combination of both (BM) on the production and health of fish. A total of 420 fish with an average initial body weight of 200 ± 0.5 g were randomly distributed in triplicate to seven diet groups (C, B50, B100, M50, M100, BM50, and BM100, respectively). The growth performance and feed utilization of fish fed with partial or total replacement levels of FM with B were not significantly affected (*p* > 0.05) during the 6 weeks of feeding. In contrast, significant differences were observed between the groups fed with a diet where FM was totally replaced with M meal and the control in terms of final body weight, specific growth rate, feed conversion ratio, protein efficiency ratio, and protein productive value. Among the blood plasma biochemistry parameters, total cholesterol exhibited a significant difference (*p* = 0.007) between the M treatments and the control diet. The fatty acid profile of the liver was changed with respect to the long-chain polyunsaturated fatty acid content in all experimental groups. Parallel with this, the upregulation of *elovl5* and *fas* genes in liver was found in all experimental groups compared to the control. Overall, this study shows that fish meal cannot be substituted with yellow mealworm meal in the practical diet of African catfish without compromising the growth, health and feed utilization parameters.

## 1. Introduction

Fish meal (FM) plays a priceless role in the success of the fish feed industry due to its high nutritional value. However, a supply shortage increases its price and leads to high feed costs [1]. It is crucial to reduce the use of FM in fish diets by replacing it with cheap and sustainable alternative protein sources because total dependence can influence the overall operations of fish-farming sectors and consequently reduce profitability [2,3]. As a result, a number of new ingredients, including plant ingredients [4], processed animal ingredients [5,6], and ingredients from microalgae [7,8], have been investigated over the last few years. Among the plant ingredients, soybean meal (SBM) is widely used due to its favourable amino acid profile and sustainable supply [5,9]. The SBM has been used as a replacement for FM in a number of freshwater species including the African catfish [10,11,12,13]. However, the inclusion of SBM at high levels can lead to poor growth and poor protein utilization [9,12,14]. Currently, attention is being directed to the possibility of using insect meals as FM substitutes in fish feed due to their relatively high nutritional quality, sustainability and scalability, less need for arable land, privileged legal backgrounds, acceptability by many fish species, and low environmental effects [15,16,17,18,19,20]. In addition, the use of insects for feeding fish represents a promising alternative because of high protein content, a balanced amino acid profile and high nutrient digestibility [21,22]. The average crude protein content of several insects varies between 50 and 80% (dry matter basis), which is almost similar to the crude protein content of the fish meal [21], which can meet the nutrient requirement of wide varieties of fish species [23]. Insects also contain some vitamins, minerals, and bioactive substances in addition to the protein and lipid components [24,25].

However, the low content of long-chain polyunsaturated fatty acids (LC-PUFA), especially eicosapentaenoic (EPA; 20:5n−3) and docosahexaenoic (DHA; 22:6n−3) acids, is one of the main nutritional limitations of including insects, especially in the diets of marine fish [26]. This is because marine fish cannot synthesize EPA and DHA from the precursor alpha-linolenic acid (18:3n−3) in their diets. While freshwater fish are capable of synthesizing EPA and DHA from dietary alpha-linolenic acid, dietary EPA and DHA are more efficiently deposited in fish meat and oil [26,27]. Fish meal is the main dietary source of n−3 LC-PUFA; thus, high FM replacement levels with insect meal should not compromise the recommended dietary levels of n−3 LC-PUFA for a given species [28,29]. In addition, insect meals have a limited phosphorus content when compared to FM and SBM and this low phosphorus content may affect lipid metabolism as well as the growth performance of fish [26]. However, the possibility of modifying the nutrient composition of insect meals by altering the substrate composition offers a great advantage to overcome the lack of LC-PUFA and other ingredients in insect meals [30,31].

Many insects have been studied in the African catfish (*Clarias gariepinus*) feed, for instance, black soldier fly (*Hermetia illucens*) [32,33,34], mealworm ((*Tenebrio molitor*) [35], common house fly maggot (*Musca domestica*) [36], crickets (*Gryllus bimaculatus*) [37,38], butterfly (*Cirina butyrospermi)* caterpillars [39,40], and grasshoppers (*Caelifera*) [41]. Due to the insect species-specific differences, the optimal dietary replacement levels of FM with insect meals vary significantly between these studies, ranging from 10% to 100% (37.5 to 760 g kg^−1^) with varying degrees of success [32,33,34,35,36,37,38,39,40,41]. The success of the replacement usually depends on the digestibility of feed ingredients. Appreciable high apparent digestibility coefficients on dry matter (72–87%) and crude protein (82–94) were demonstrated for defatted-B-meal-containing diets in the case of several aquaculture species [42,43,44], and a similarly high value was reported for partially defatted M meal in rainbow trout [45]. The differences in digestibility of the insects might be related to the chitin content of the insect’s meal, and the developmental stage of the harvested insects [46], as well as to the drying process of the ingredient (spray vs. oven dry) [42]. Dietary chitin may impair the digestibility of other nutrients when it is greater or equal to 10% [47]. According to these advantages and limitations, it is apparent that insect’s meal could play a role in the future diets of African catfish.

The above-highlighted studies were so far carried out on small-sized (2.4 to 10 g initial weight) African catfish, and there are scarce studies on bigger-sized fish (≥200 g). Additionally, the evaluation of black soldier fly meal and mealworm meal inclusion at the same time has not been studied for this fish species. Previously, the positive effect of the simultaneous inclusion of three insect oils (black soldier fly, mealworm, and silkworm pupae) was reported on growth performance, lipid metabolism and the inflammatory response of juvenile mirror carp by Xu et al. [48]. Thus, the objective of this study was to investigate the effects of black soldier fly and mealworm meals on the production performance, feed utilization, lipid metabolism, and immune system of African catfish growers.

## 2. Materials and Methods

### 2.1. Experimental Design and Rearing Conditions

A six-week experiment was carried out at the Hungarian University of Agriculture and Life Sciences, Kaposvári Campus, at the Department of Aquaculture, using a recirculation aquaculture system (RAS). A complete randomized design (CRD) was used to set up the experiment. A total of 420 African catfish individuals with an average initial weight of 200 ± 0.51 g were randomly distributed in 21 tanks in triplicate (20 fish per tank) and acclimatized for one week. Water parameters (temperature, NH_4_^+^, NO_3_^−^, NO_2_^−^, dissolved oxygen, and pH) were checked regularly during the trial, and the average values were as follows: temperature: 24.5 ± 0.2 °C; dissolved oxygen: 4.2 ± 0.5 mg L^−1^ measured daily; NH_4_^+^: 0.50 ± 0.02 mg L^−1^, NO_3_^−^: 28.5 ± 0.19 mg L^−1^, NO_2_^−^: 0.16 ± 0.02 mg L^−l^, and pH: 7.1 ± 0.2 measured on a two-weekly basis.

### 2.2. Ethical Issues

All procedures involving fish were conducted in line with the Hungarian legislation on experimental animals and approved by the National Scientific Ethical Committee on Animal Experimentation (identification number of the license: KA-3403). All efforts were made to minimize the fish’s suffering. For that reason, the fish were anesthetized before being sacrificed using Norcaicum-based anesthetics (50 mL 100 L^−1^).

### 2.3. Feed Preparations and Feeding

Seven experimental diets were formulated as follows: The control (C) diet contained 200 g kg^−1^ of fish meal (FM); in the experimental diets (B50, B100, M50, M100, BM50, BM100), FM was replaced partially (50%) and totally (100%) with different insect meals (Figure 1). In diet M50, 100 g of FM was replaced with mealworm meal (M); in diet M100, 200 g of FM was replaced with M; in diet B50, 100 g of FM was replaced with black soldier fly meal (B); in diet B100, 200 g of FM was replaced with B; in diet BM50, 100 g of FM was replaced with a 1:1 combination of M and B; and in diet BM100, 200 g of FM was replaced with a 1:1 combination of M and B. The chemical composition of insects used in feeds is similar to data presented by Sandor et al. [49]. The experimental diets were set to be iso-nitrogenous and iso-energetic (Table 1). The feed ingredients were thoroughly mixed to form a homogenous blend, moistened with water (200 mL kg^−1^), and then extruded using a single screw extruder (Abrazive, Hungary) to produce 6 mm sinking pellets, which were dried in an oven at 55 °C (Pol-Eko, Wodzislaw Slaski, Poland).

Crude protein varied between 42.9 and 46.5% on a wet weight basis in the test and control diets (Table 2). Crude lipid and caloric contents were in the range of 8.0–9.3%, and 19.1–20.0 KJg^-1^, respectively. Table 3 shows the main fatty acid and amino acid compositions of the experimental feeds. We hypothesized that with FM replacement, a considerable decrease in the EAA would be inevitable. Therefore, to prevent the EAA deficiency, some supplementation with premix containing lysine and methionine was used in the feed, according to the formulation. Based on the measurements, the sum of essential amino acids (ƩEAA) varied between 18.61 and 24.0%. The levels of the main limiting essential amino acids, lysine and methionine, were adequate for catfish [23], ranging from 2.59 to 4.01% and 1.00 to 1.22%, respectively. The M100, BM50, and BM100 groups had a higher lysine content compared to others, while an increase in tryptophan content was observed with B-meal inclusion (B50 and B100 groups). In terms of the fatty acid composition, an increase in lauric acid (12:0) levels was observed when FM was replaced with B-meal. Similarly, a decrease in stearic acid (16:0) level with the addition of insect meals was detected. The B100 diet presented the highest level of total saturated fatty acids (due to the highest levels of 12:0 and 14:0, respectively). M100 diets contained the highest levels of MUFA (due to the highest levels of 18:1n−9). With regard to Lc-PUFA levels, the control diet presented the highest total level (9.20%) with 2.34% EPA, 6.00% DHA, and 0.85% ARA content, while the test diets showed lower total levels of 7.03% (B50), 8.05% (B100); 7.23% (M50), 5.37% (M100), 7.01% (BM50), and 6.51% (BM100), respectively.

The feed was fixed at 2% of biomass during the six weeks of the experiment. The tank biomass was measured to adjust the daily feed portions every second week with a portable measuring scale (accuracy ± 1 g). The daily feed amount was distributed by hand five times per day according to the fish appetite. After daily feeding, the leftover feed was monitored via siphoning when a significant number of uneaten particles were not observed in any treatments. Accordingly, we considered that all given feed was taken up and utilized by fish.

### 2.4. Chemical Analysis

The chemical composition of feeds (Table 2) and feces was analyzed using standard methods of the AOAC [50]. Crude protein (CP) was determined with the Kjeldahl method (AOAC 928.08) using a digestion block (KJELDATHERM, Gerhardt, Germany) via a distillation procedure (VAPODEST 450, Gerhardt, Germany). Approximately 0.5 g of dry samples was digested with 10 mL of 98% H_2_SO_4_ and 10 mL of 30% H_2_O_2_; afterwards, the generated ammonium sulphate was distilled off using 2% H_3_BO_3_. The CP was calculated as N × 6.25 for diets and feces. The crude fat was determined from 5 g of dry diet samples according to the AOAC 945.16 Soxhlet method using an automatic system (SOXTHERM^®^ Unit SOX416, Gerhardt, Germany) and diethyl ether (boiling point, 40–60 °C) as a solvent. The crude ash content of the diets was estimated according to the AOAC 942.05 method. Two grams of the samples was weighed and placed in a furnace heated to 550 °C and held for 4 h. The amount of ash that remained was recorded. Crude fiber content of the diets was determined from defatted samples (AOAC 928.08). The sample amount was 1.5–2.0 g, and the digestion procedure was carried out using 0.13 M H_2_SO_4_ and 0.31 M NaOH in a GERHARDT Fibretherm FT12 apparatus (Königswinter, Germany). The acid-dissolved fiber (ADF) was determined with the same equipment by using an ADF solution prepared from N-acetyl-trimethyl-ammonium bromide dissolved in 0.5 M H_2_SO_4_ and a few drops of antifoaming agent. The chitin content was determined as the difference between ash-free acid dissolved fiber (ADF) and protein linked to ADF (ADIP) (chitin% = ADF%-ADIP%) according to Finke [51] and Marono et al. [52]. The gross energy of the diets was determined using a Parr 6400 Automatic Isoperibol Calorimeter (Parr Instrument Co., Moline, IL, USA) calibrated with benzoic acid. 

The amino acid content of the diets (Table 3) was analyzed using the UPLC-DAD method (Waters Acquity UPLC H-Class, Milford, CT, USA) after acid hydrolysis and pre-column derivatization with the 6-aminoquinolyl-N-hydroxysuccinimidyl carbamate (AQC) reagent. The analysis was performed with AccQ UPLC BEH C18 (2.1 × 100 mm, 1.7 μm) column (Waters) and AccQ Tag Ultra eluents A, B and water in gradient mode, and the flow rate was 0.7 mL min^−1^. The chromatograms were evaluated at 260 nm, using amino acid standards. Acid hydrolysis was carried out for amino acid analysis. Twenty-five milligrams of the samples were hydrolyzed by 6 N HCl containing 1% of phenol in a Milestone Ethos One microwave digestion system. Hydrolysates were completed to 5 mL with 1 M borate buffer (pH 8.51).

The calcium and phosphorus content were analyzed using the ICP method. The digestion of samples was achieved with mixtures of acids, including nitric acid (R.G. 65%) and hydrogen peroxide (R.G. 30%). The extraction was realized using microwave digestion technique under high pressure and a Milestone Ethos Plus (Sorisole, Italy) microwave apparatus. The concentrations of elements were measured using Thermo Scientific 6500 ICP-OES (Waltham, MA, USA) equipment. 

Fatty acids were analyzed via gas chromatography (Agilent 7890A GC System) as methyl esters of fatty acids [53]. Ten milligrams of fat extracted with diethyl-ether was saponified with 250 μL of 0.5 M KOH/methanol in a 2 mL screw-cap glass vial and heated for 15 min in a block thermostat at 140 °C. The transesterification was performed with boron trifluoride in methanol (14%, Sigma, Inc., St. Louis, MO, USA) for 10 min, and then 250 μL of heptane (Merck, Darmstadt, Germany) was added, and the mixture was brought to a boil. After cooling, saturated sodium chloride solution was added up to 2 mL. After standing for 1 h, the separated upper phase was moved into a test vial containing a 1 mm layer of dry sodium-sulfate, to which 1 mL of heptane was added. The column was a Supelco SP-2560 with film dimensions of 100 m × 0.25 mm × 0.2 μm (Supelco, Bellefonte, PA, USA). The oven temperature program was 5 min at 140 °C, which was then increased by 4 °C/min until 240 °C, with 10 min for the final temperature. The injector temperature was 220 °C, and the detector temperature was 250 °C. The carrier gas was hydrogen, with a column flow of 1 mL min^–1^ and an automatic injection amount of 1 μL.

### 2.5. Samples Collection

Body weight, total length and liver weight of the fish were measured individually at the end of trial. For growth parameters, all (60 fish per diet group or 20 fish per tank) were individually measured. A total of 18 fish per diet group (6 individuals per tank) were randomly taken and dissected for somatic indices. Liver samples from two individuals per tank were frozen at −80 °C for fatty acid analysis. Another three fish per tank were collected for whole-body composition analysis, and five fish samples were obtained before the beginning of the trial.

Blood samples were collected from the caudal veins of fish using heparinized 1 mL syringes, needles and 1.5 mL microcentrifuge tubes (3 fish per tank or 9 fish per treatment). Plasma was separated through centrifugation (CAPP CR-1730R, Nordhausen, Germany) at 4 °C and 1700× *g* for 20 min and stored at −20 °C for further analysis. Liver samples were taken from three fish per tank for gene expression analysis. For each sample, 100 mg of liver was collected and placed in 1 mL of RNA later (Invitrogen, Thermo Fisher Scientific, Waltham, MA, USA) for one day at 4 °C, followed by storage at −20 °C until analysis.

### 2.6. Blood Plasma Biochemistry

The total protein (TP) concentration of plasma was determined via a colorimetric assay based on the biuret reaction using a protein diagnostical reagent kit (Fluitest TP, Analyticon Biotechnologies AG, Lichtenfels, Germany), according to the manufacturer’s instructions.

To determine the total immunoglobulin (Ig) level of plasma samples, 50 µL of plasma and 50 µL of polyethylene glycol (PEG) were added to each well of a 96-well microtiter plate. After two hours of incubation at room temperature, plates were centrifuged at 1000× *g* for 15 min. Total protein contents of the supernatants were measured using FLUITEST Total Protein Kits (Analyticon Biotechnologies AG, Lichtenfels, Germany). These values were subtracted from the total protein levels of the samples, which had been measured previously. The result was equal to the total immunoglobulin concentration of plasma.

Alkaline phosphatase (ALP), alanine aminotransferase (ALT), amylase (AMY), creatine (CREA), calcium (CA), total cholesterol (CHOL), glucose (GLU), globulin (GLOB) and phosphate (PHOS) levels of plasma samples were measured using a Samsung PT10V blood analyzer and Comprehensive Plus test assays (Samsung, Seoul, Republic of Korea).

### 2.7. Gene Expression

Expression levels of genes involved in fatty acid metabolism (fatty acid synthase (*fas*), fatty acid desaturase 2 (*fads2*) and elongases (*elovl2* and *elovl5*) were measured in liver samples through real-time quantitative PCR (qPCR), using elongation factor-1 alpha (*elf1α*) as an internal reference gene. Total RNA was isolated using the SV Total RNA Isolation System (Promega, Madison, WI, USA) according to the manufacturer’s instructions. The quantity of the RNA was measured using a NanoDrop 2000 spectrophotometer (Thermo Fisher Scientific, Waltham, MA, USA). The integrity (quality) was checked by denaturing gel electrophoresis (1.5% agarose gel) and the purity by measuring the OD_260_/OD_280_ absorbance ratio (>1.95). The cDNA was generated from 400 ng of total RNA using a qPCR cDNA Synthesis Kit (PCR Biosystems, London, UK) following the manufacturer’s protocol. The product of the first-strand cDNA synthesis was stored at −20 °C until the quantitative RT-PCR (qPCR) runs. The qPCR reactions were carried out using a LightCycler 96 instrument and the FastStart Essential DNA Green Master qRT-PCR kit (Roche, Switzerland). The primers used are presented in Table 4. The qPCR reaction was carried out in a final volume of 20 µL consisting of 10 µL of master mix (2×), 1 µL of each primer (10 µM), 5 µL of cDNA (reverse transcription reaction mix) and 3 µL of nuclease-free water. The thermal profile for all reactions was 95 °C for 10 min, followed by 45 cycles at 95 °C for 15 s, 60 °C for 30 s and 72 °C for 30 s. The specificity of the reactions was checked via melting curve analysis, and no mispriming or primer dimers were found. All reactions were carried out in triplicate. The mean threshold cycle (Ct) values were calculated, and the qPCR data were analysed using the method described in [54]. The efficiencies of qPCR reactions were determined using standard curves, and serial dilutions were performed from cDNAs of a liver sample. These cDNAs were diluted to 1×, 30×, 90×, 270×, and 810×. Quantitative PCR reactions were carried out on these dilutions with all primer pairs in triplicates. Standard curves were drawn for each primer pair by plotting Ct values against the log_10_ of different dilutions of cDNA sample solutions. Efficiencies (E) were calculated from the slopes of the standard curves using the equation E = 10(−1/slope), and the results are shown in Table 4.

### 2.8. Calculations and Growth Models

In order to determine the growth performance and nutrient utilization of the fish, the following parameters were measured and calculated at the end of the trial:specific growth rate (SGR), SGR = 100 × (lnwf − lnwi)/t;weight gain (WG) = Wf − Wi;protein efficiency ratio (PER) = weight gain (g)/protein intake (g);feed conversion ratio (FCR) = feed intake (g)/weight gain (g);protein production value (PPV %) = 100 × (final protein content of fish biomass – initial protein content of fish biomass)/protein intake;where wf = final mean weight (g); wi = initial mean weight (g); t = experimental time; in days; Wf = total final biomass; Wi = total initial biomass;survival rate (S %) = 100 × number of survived fish/numbers of stocked fish;biometric indices and post-harvest indices determined were: Hepatosomatic Index (HSI%) = LW/BW × 100; LW = liver weight (g), BW = body weight (g); Condition factor (CF) = Total BW/TL3 × 100 and TL = total body length (cm).

In order to reveal whether there were significant differences in growth patterns between treatments, a linear and an exponential growth model were fitted on individual weight data recorded at bi-weekly measurements. Equations (1) and (2) were parameterized with the use of interaction terms between time and treatment dummies among the predictors to assess the impact of diet on growth rates:w_t_ = w_in._ + β_1_ t + β_2_ B_50_ t + β_3_ B_100_ t + β_4_ M_50_ t + β_5_ M_100_ t + β_6_ B M_50_ t + β_7_ BM_100_ t(1)
w_t_ = A exp (κ_1_ t + κ_2_ B_50_ t + κ_3_ B_100_ t + κ_4_ M_50_ t + κ_5_ M_100_ t + κ_6_ BM_50_ t + κ_7_ BM_100_ t)(2)
where w_t_, the observed mean weight (g) of fish at time t; w_in._ and A, the initial weight estimates (which would only be accepted if it was ~200 g); t, the time (days) since the initial state; B_50_, B_100_, …, BM_100_, the dummy (0 or 1) variables representing the treatments other than control group; exp (), the base of the natural logarithm; β_i = 1,2,…,7_, the parameters to estimate representing the daily weight gain (g/day), out which β_1_ is the estimated weight gain in the control group, while β_2_, β_3_, …, β_7_ are the additional weight gain parameters associated with their respective diets; κ_i = 1,2,…,7_, the parameters to estimate representing specific growth rate (%/day), out which κ_1_ is the estimated growth rate in the control group, while κ_2_, κ_3_, …, κ_7_ are the additional growth rates in each group. 

After linearizing the exponential growth function (Equation (2)) by taking the natural logarithm of both sides of the equation, the estimates were obtained via ordinary least squares. 

### 2.9. Statistical Analysis

Data were first analyzed using one-way ANOVA, except for gene expression data. The means between groups were compared using post hoc Tukey’s multiple comparison test. Significant differences were considered for a *p*-value < 0.05. The Shapiro–Wilks test was used to assess normality. The homogeneity of the variances was checked with Levene’s test. Polynomial contrasts (linear and quadratic response) were used to determine if there were linear or quadratic patterns in the relationship between main estimated indicators and FM replacement level (0, 50, and 100%) with insect protein in B, M and BM diets. For these analyses, three comparison groups were formed:Comparison B, between control, B50, and B100;Comparison M, between control, M50, and M100;Comparison BM, between control, BM50, and BM100.

Relationship between crude fat content of liver and HSI was explored with the computing Pearson correlation. All the statistical analyses were performed using IBM SPSS Statistics 27 software. Gene expression data were evaluated using the Relative Expression Software Tool (REST ©) [57], where each group was individually compared to the control.

## 3. Results

### 3.1. Growth Performance and Feed Utilization Efficiency

Replacement of fish meal with black soldier fly meal (B) did not compromise feed/protein utilization (PER, PPV, FCR) and the growth of fish (FBW, SGR), as among the B-diet treatments (Comparison B), there were no significant differences in these performance indicators. On the other hand, inclusion of yellow mealworm into the catfish feeds (M and BM diets) had a negative impact on growth and feed conversion (Table 5). The results of the polynomial contrast analysis support a statistically significant linear relationship between production parameters and the level of fish meal replacement in M and BM diets (Comparison M and BM). These linear patterns are illustrated for SGR and FCR in Figure 2. Protein intake (PER and PPV) was negatively affected only by the M100 diet (*p* = 0.007, *p* = 0.002, respectively). Within the biometric indices, significant differences between the control group and treatments were not found in any of the comparison groups (Table 6). High biological variations were observed in HSI within the same treatment. The survival rates for C, B50, B100, M50, M100, BM50, and BM100 treatments were 90.0, 100, 90.0, 91.6, 90.0, 91.6, and 93.3%, respectively. Since the number of fish per tank was 20 and a survival rate of 90% corresponds to a mortality of two fish per tank, we did not analyze the difference between survival rates statistically.

Data from fitted growth models are presented in the Appendix A (Table A1 and Table A2) Parameter values and their significance levels in both the linear and exponential growth functions indicate that treatments M100 and BM100 would reduce the growth of catfish in comparison with the control group over a longer time horizon. Although coefficients for B50, B100, M50, and BM50 dummies are also negative, contrary to M100 and BM100, these are not significant at the *p* < 0.05 level. Figure 3 illustrates fish growth patterns under different treatments simulated beyond the timeframe of the current experiment.

### 3.2. Biochemical Parameters

As shown in Table 7, most of the plasma biochemical parameters analyzed were not significantly influenced by the diet, except for cholesterol, globulin, and phosphate. The cholesterol levels increased significantly in fish fed with different inclusion levels of M (M diets *p* < 0.001, BM diets *p* = 0.006). Phosphate levels decreased significantly in the B50, M50, and M100 treatments compared to the control group. Globulin level was increased in all experimental treatments, significantly so in BM groups (*p* = 0.002). The following concentration levels were measured: ALP (27–53 U L^−1^), ALT (10–76 U L^−1^), calcium (10.9–16 mg dL^−1^), cholesterol (80–176 mg dL^−1^), creatine (0.1–0.4 mg dL^−1^), globulin (2.1–3.6 g dL^−1^), glucose (57–135 mg dL^−1^), gamma-glutamyl transpeptidase (GGT) (<5), total protein (2.3–3.7 g dL^−1^), phosphate (4.7–9.6 mg dL^−1^), immunoglobulin (0.19–0.26 mg mL^−1^) and amylase activity (10–44 U L^−1^).

### 3.3. Total Lipid Content and Fatty Acid Composition of the Liver

As shown in Table 8, the total lipid fat content presented numerical variability between the fish belonging to different diet groups. However, the difference was not statistically proven (*p > 0.05*). Replacement of FM with B meal caused the appearance of lauric acid (12:0) in the B50, B100, and MB100 diets. In the case of myristic acid (14:0), a significant difference was found between the B100 and control group (*p* = 0.018). Total SFA was the highest in groups B50 and B100, but without a significant difference compared to the control group. The level of DHA (22:6n−3) was significantly (*p* = 0.011) reduced when FM was replaced with BM (BM comparison). Significant differences (*p* < 0.001) in Lc-PUFA between C and the rest of the treatments were detected. The n−6/n−3 ratio was lowest in the control group and significantly different from B100, M50, M100, BM50, and BM100 treatments. A negative correlation (Pearson R = −0.308, *p* = 0.047, N = 42) was found between HSI and total lipid content of the liver (Table 9).

### 3.4. Gene Expression

Compared to the C group, the relative expression of the *elovl5* and *fas* genes was significantly (*p* < 0.05) higher in the B100 and M50 groups, respectively. No significant differences were found in the cases of other examined genes and groups. However, all examined genes exhibited a similar pattern with respect to the B (B50 and B100) and M (M50 and M100) diets (Figure 4). With respect to the BM meal, the expression of *fas*, *fads2*, and *elovl2* was higher in fish fed with BM100 diet than in fish fed with BM50 diet, while that of *elovl5* showed an opposite pattern, being higher in fish fed with BM50 than in fish fed with BM100, with no significant differences. In addition, *elovl5 and fas* were upregulated in all fish fed the different experimental diets compared to the control diet, while *elovl2* and *fads2* indicated a downregulation in fish fed the M100 and BM50 diets, respectively.

## 4. Discussion

In the present study, black soldier fly and yellow mealworm’s different inclusion levels and their combination in the diets of African catfish grower were tested. In order to ensure the nutrient requirement of the diets following the replacement of fish meal with different insects, essential amino acid supplements were added. Otherwise, the different AA composition of the insects would cause a deficit in the aquafeeds. As a consequence of this, the level of the limiting AA in our diets was above the optimum recommended level. The results of the feeding trial revealed that there were no significant (*p* > 0.05) differences between the B treatment groups in terms of growth, weight gain, and nutrient intake. This finding agrees with the findings reported on Siberian sturgeon (*Acipenser baerii*) fed with B meal [58], Nile tilapia (*Oreochromis niloticus*) fed with different inclusion levels of B [59], and zebrafish (*Danio rerio*) fed with B [60]. Similarly, Melenchón et al. [61] reported the absence of a negative effect on the growth performance of rainbow trout (*Oncorhynchus mykiss*) fed with fish feeds where fish meal was partially and totally (130 and 300 g kg^−1^) replaced with B and M meal, or in Jian carp *(Cyprinus carpio var. Jian)* [62] and in common carp *(Cyprinus carpio)* [63], using feeds with 140 g kg^−1^ and 120 g kg^−1^ B protein levels, with 100% FM replacement without having a significant effect on growth. In contrast to the B meal, our diets with M replacement (M100, BM100) significantly inhibited the growth performance and nutrient utilization of catfish growers, while partial replacement of FM (M50, B50, and BM50) exhibited approximately similar parameters to the control diet. These findings are comparable with previous findings in the literature. Partial inclusion levels of FM with B and M showed the best results in African catfish with an initial body weight of 2.7, 6.0, and 5.1 g [32,35,64]. With the application of growth models generated using data from the feeding period, total replacement of FM with BM would reduce the time involved in the growth of catfish compared to the control group, as was already observed in the case of M meal. The decreasing tendency observed in our case in the growth performance of African catfish as the replacement level of FM with M meals increased could be associated with the lower palatability or poor digestibility of insects compared to FM. This is in line with previous findings presented by Sandor et al. [49], when apparent digestibility coefficients (ADC) were calculated. In that study, low ADC amino acid was calculated for M meal fed to African catfish. In spite of that, the EAA levels of our diets containing M meal (Table 3. M100, BM50, BM100) were higher compared to the rest of the diets, due to its low digestibility, as M meal obstructed the growth performance of fish and protein efficiency of diet. In contrast with our findings, no significant effect on the growth of European sea bass *(Dicentrarchus labrax*) fed with different inclusion levels of defatted M meal [65] was reported. The variations in the results might be associated with differences in doses (g kg^−1^) of insect meals and the presence of other ingredients in the formulated diet, age and size of fish [45], the feeding habits of fish [3], processing (either full fat or mechanically defatted), and the high lipid content that reduces the availability of crude protein [66]. Thus, for the commercialization of insect-based fish feeds, reliable ingredients need to be used, with information regarding its digestibility and palatability. 

The condition factor (CF) of a fish reflects physical and biological situations and fluctuations caused by interactions among feeding conditions, parasitic infections, and physiological factors [67]. In the present study, the condition factor of African catfish was not significantly (*p* > 0.05) affected by dietary treatment. This might indicate the situation with fish exhibiting isometric growth (presence of length increases in equal proportion with body weight) [68]. In the present study, 100% survival was observed in the B50 group, while some mortalities occurred in the other groups. The mortality was related more to handling stress than the feed. However, the positive impact of B50 replacement in coping with stress needs further investigation. The quality and utilization of protein in a diet is measured through the protein efficiency ratio (PER) and the protein productive value (PPV) [69]. The latter is also known as protein retention and is represented through the ratio between the protein retained in fish tissues and the dietary protein fed [70]. In the present study, feed utilization did not significantly (*p* > 0.05) differ between the B treatments. Nevertheless, the highest result was obtained with the control diet. In contrast, when using M meal in the diets, the protein efficiency was significantly decreased when FM was totally replaced with M or close to significant level (*p* < 0.05) using BM meal. Higher feed utilization of the control group might be associated with the lowest dietary chitin and fiber pattern. It was confirmed that the growth performance, FCR, and SGR of Nile tilapia decreased as the dietary chitin increased [71]. Chitin content higher than 10% decreased the growth performance, and up to 5% increased the growth of Nile tilapia [72]. The other reason might be associated with high calcium and phosphorous levels [73], which is the case in our control diet as well. 

Plasma glucose and total protein levels are associated with the stress response [74]. Interestingly, despite the numerical differences, the values of plasma glucose (GLU) were not statistically significant (*p* > 0.05) between the treated and control diet groups. On this note, a previous report showed the absence of a significant difference (*p* > 0.05) in GLU concentration in juvenile black porgy (*Acanthopagrus schlegelii)* fed with M meal as FM substitution [75]. The plasma phosphate level was significantly (*p* < 0.05) decreased with dietary insect meal inclusion, showing a lower available P in insects compared to FM.

High levels of energetic metabolites such as cholesterol can be indicators of liver pathology and a high-fat diet [76]. In the present study, the cholesterol level of blood plasma was significantly lower in the control compared to M100, BM50, and BM100. The diets were formulated to be isolipidic with an average of 10% fat content, which excludes the possibility of causing elevated plasma cholesterol levels. This result is in contrast with the findings of Tran et al. [77] and Jeong et al. [78], who reported that defatted mealworm meal did not significantly affect the serum cholesterol level of European perch (*Perca fluviatilis*) and rainbow trout fed with diets containing 20.3% and 28% dietary protein (FM) replacement with defatted and fully fatted M, respectively. Our results suggest that replacing FM with defatted M meal may cause a higher cholesterol level in the blood and an increased hepatosomatic index (HSI%) in the fish (Table 6). However, the HSI did not differ significantly with FM replacement. A similar observation was found in the fat content of the liver, which seems to increase with FM replacement level, but a significant difference could not be detected. Indeed, the level of cholesterol was below 200 mg dL^−1^, indicating it was within the normal range observed in the case of fish [79]. The negative correlation observed between the fat content of the liver and the HSI (Table 9) would indicate that the fish were healthy without signs of hepatosteatosis.

Regarding the fatty acid composition of liver, the Lc-PUFA deficit could be observed due to the replacement of FM in the experimental diets. At the same time, the excess in SFA in B treatments and the increase in the n−6/n−3 ratio in all experimental groups reflected dietary FA composition. Generally, when fish are fed low levels of n−3 Lc-PUFA, *fads2* expression is upregulated in the liver, the tissue of which is the most active in terms of lipid metabolism and Lc-PUFA biosynthesis in fish. In our study, *fads2* expression in fish liver fed with diets with 100% replacement was higher compared to 50% replacement in the case of black soldier fly meal, which are findings similar to those presented in [80]. Previous studies have shown that vertebrates possess three members of the Elovl protein family with roles in PUFA elongation, namely Elovl2, Elovl4 and Elovl5, that differ in their fatty acid (FA) substrate specificities [81]. Elovl5 has a preference for C18 and C20 PUFA, whereas Elovl2 is predominantly involved in elongation of C20 and C22 PUFA. Elovl4 participates in the elongation of very long-chain (C ≥ 24) PUFA substrates found in the retina and testis. In the present study, the relative expression of genes encoding Elovl2 (*elovl2*) and Elovl5 (*elovl5*) proteins were investigated, and positive responses of *elovl2* in experimental groups were found. A similar trend is observed for *elovl5*. Based on this finding, a slight increase in endogenous Lc-PUFA biosynthesis capacity was observed in all treatments, irrespective of the type of insect, in order to compensate for the reduction in dietary Lc-PUFA that occurred with the replacement of fish meal. 

## 5. Conclusions

In conclusion, the growth performance and feed utilization of African catfish fed with partial or total replacement levels of fish meal with B were not significantly affected (*p* > 0.05) during 6 weeks of feeding. In contrast, inclusion of yellow mealworm in the catfish feeds (M and BM diets) had a negative impact on growth and feed conversion. Cholesterol levels in blood serum and the hepatosomatic index of fish were negatively influenced by the inclusion of M meal, but the total lipid content of the liver remained unchanged. The Lc-PUFA content of the liver in experimental fish was generally decreased, and a slight increase in the endogenous Lc-PUFA biosynthesis capacity was observed in all treatments, irrespective of the type of insect. Based on this finding, B can successfully replace FM (200 g kg^−1^) partially or totally in the practical diet of African catfish without having a significant adverse effect on health, nutrient utilization, and growth performance. As regards M dietary inclusion, this ingredient may hinder fish growth and cause health problems for this fish species when a high inclusion level is applied. 

## Figures and Tables

**Figure 1 animals-13-00968-f001:**
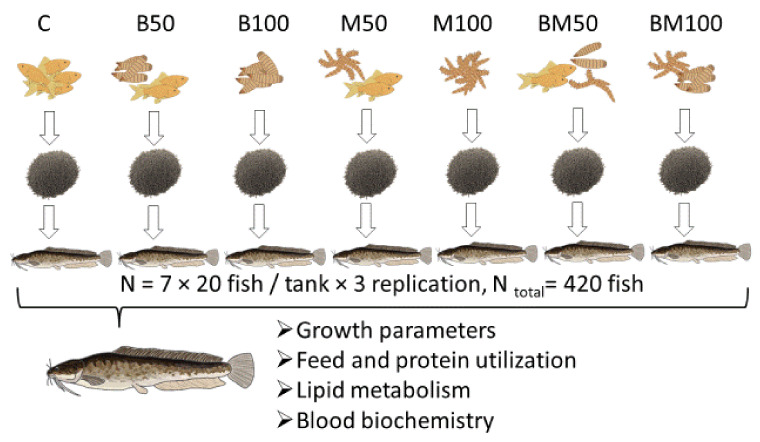
Schematic diagram of the experimental setup and the measured parameters at the end of the trial.

**Figure 2 animals-13-00968-f002:**
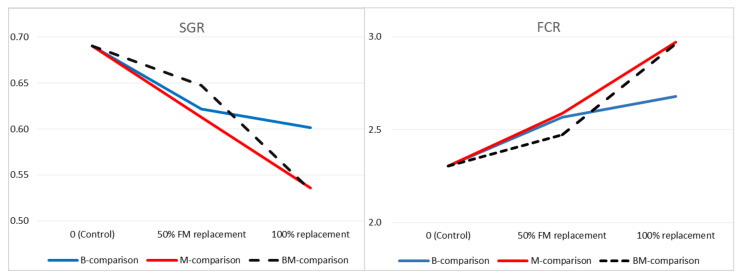
Variation in SGR and FCR parameters according to inclusion level of insects.

**Figure 3 animals-13-00968-f003:**
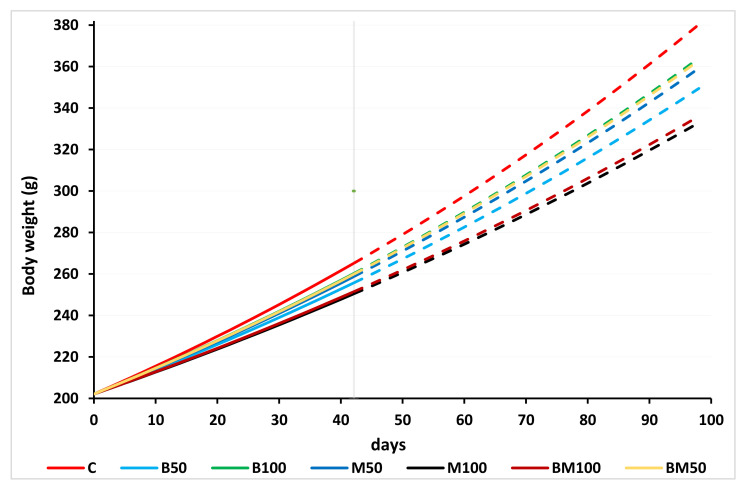
Simulation of growth of fish fed under different treatments using the parameterized exponential growth function (Table A2).

**Figure 4 animals-13-00968-f004:**
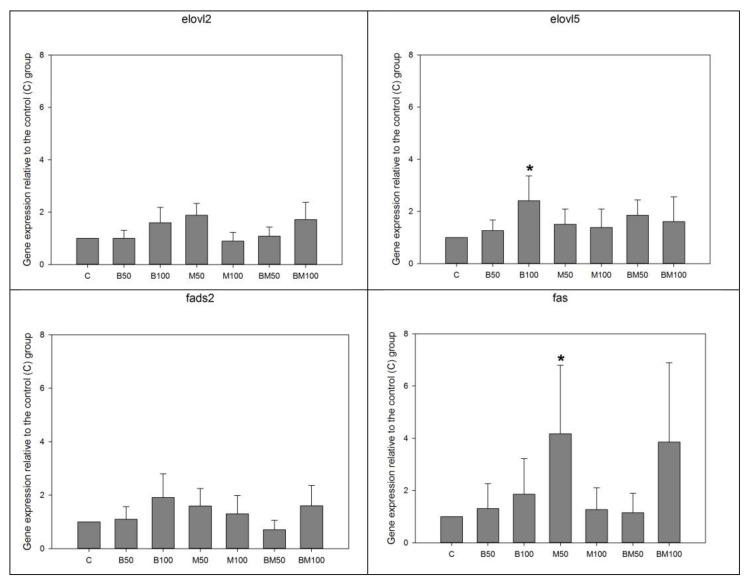
Nutritional regulation of fatty acyl desaturase (*fads2*), fatty acyl elongases 2 (*elovl2*) and 5 (*elovl5)*, and fatty acid synthase (*fas*) in liver of African catfish grower fed a control diet and six different experimental diets. Values are normalized expression ratios relative to the control (C) diet. *: significant (*p* < 0.05) difference compared to the control (C) group.

**Table 1 animals-13-00968-t001:** Formulation of the diets with different inclusion level of insect meal (%).

Ingredients	C	B50	B100	M50	M100	BM50	BM100
**Fish meal (FM) ^1^**	20	10	0	10	0	10	0
**Mealworm (M) ^2^**	0	0	0	10	20	5	10
**Black soldier fly (B) ^3^**	0	10	20	0	0	5	10
**Soy protein conc. ^4^**	14.6	14.6	14.6	14.6	14.6	14.6	14.6
**Wheat**	33.5	33.2	33.0	32.7	31.8	33.0	32.4
**Poultry meal ^5^**	25	25	25	25	25	25	25
**Premix ^6^**	1.5	1.5	1.5	1.5	1.5	1.5	1.5
**Rapeseed oil ^7^**	4	4.3	4.5	4.8	5.7	4.5	5.1
**Calcium phosphate**	1	1	1	1	1	1	1

C control, B black soldier fly, M mealworm, BM mixture of black soldier fly and mealworm, Premix: vitamins, minerals, lysine, methionine. ^1^ Euro-protein Ltd., Verőce, Hungary. ^2^ Berg and Schmidt Pte. Ltd., Singapore. ^3^ Agroloop Ltd., Rotterdam, The Netherlands. ^4^ Sojaprotein, Becej, Serbia. ^5^ Euro-protein Ltd., Verőce, Hungary. ^6^ Cargill Ltd., Budapest, Hungary. ^7^ Bunge Hungary Co., Martfű, Hungary.

**Table 2 animals-13-00968-t002:** Proximate composition (% wet weight basis) of the diets with different inclusion level of insect meal (%).

Treatments	C	B50	B100	M50	M100	BM50	BM100
**Dry matter**	95.78	97.22	96.58	96.37	95.73	97.10	93.74
**Crude protein**	44.35	45.53	44.48	45.11	46.46	45.97	42.90
**Crude fat**	7.95	8.45	8.85	8.29	9.03	8.50	9.31
**Crude fiber**	1.70	3.27	4.71	2.97	2.93	2.95	2.98
**Crude ash**	10.63	9.38	8.00	8.97	7.36	8.99	7.22
**Gross energy (KJ g^−1^)**	19.05	19.94	19.96	19.72	20.02	19.87	19.56
**Phosphorus**	1.47	1.33	1.13	1.28	1.04	1.24	0.95
**Calcium**	2.10	1.79	1.46	1.79	1.42	1.62	1.29
**Chitin**	4.96	5.98	6.32	6.42	7.24	5.91	5.55
**ADF**	7.69	8.62	10.14	10.25	13.43	10.31	10.62

ADF: acid detergent fiber.

**Table 3 animals-13-00968-t003:** Amino acid and fatty acid composition of the diets (% wet weight basis).

Treatments	C	B50	B100	M50	M100	BM50	BM100
*Essential Amino Acid (EAA)*							
**Arginine**	2.73 ± 0.07	2.53 ± 0.16	2.73 ± 0.11	2.60 ± 0.10	3.16 ± 0.05	2.98 ± 0.34	2.90 ± 0.11
**Histidine**	1.10 ± 0.01	1.00 ± 0.06	1.09 ± 0.06	0.88 ± 0.05	0.93 ± 0.01	1.06 ± 0.06	0.97 ± 0.04
**Isoleucine**	2.02 ± 0.06	1.81 ± 0.16	2.11 ± 0.04	1.88 ± 0.02	2.27 ± 0.07	2.43 ± 0.27	2.25 ± 0.12
**Leucine**	3.56 ± 0.11	3.17 ± 0.27	3.64 ± 0.06	3.50 ± 0.01	4.39 ± 0.21	4.39 ± 0.50	4.08 ± 0.22
**Lysine**	3.06 ± 0.11	2.59 ± 0.34	2.96 ± 0.01	2.83 ± 0.23	3.51 ± 0.36	4.01 ± 0.44	3.38 ± 0.14
**Methionine**	1.11 ± 0.02	1.01 ± 0.08	1.03 ± 0.02	1.00 ± 0.01	1.16 ± 0.03	1.22 ± 0.06	1.09 ± 0.03
**Phenylalanine**	2.06 ± 0.04	2.00 ± 0.07	2.19 ± 0.09	1.96 ± 0.14	2.27 ± 0.05	2.21 ± 0.16	2.21 ± 0.10
**Threonine**	2.00 ± 0.05	1.80 ± 0.14	2.00 ± 0.05	1.85 ± 0.05	2.23 ± 0.01	2.27 ± 0.23	2.14 ± 0.11
**Tryptophan**	0.22 ± 0.01	0.27 ± 0.00	0.43 ± 0.02	0.26 ± 0.12	0.15 ± 0.01	0.17 ± 0.02	0.15 ± 0.01
**Valine**	2.60 ± 0.09	2.43 ± 0.21	2.87 ± 0.05	2.53 ± 0.01	3.15 ± 0.11	3.26 ± 0.39	3.07 ± 0.16
ƩEAA	20.45	18.61	21.06	19.28	23.24	24.00	22.23
** *Non-Essential Amino Acid* **							
**Alanine**	2.90 ± 0.10	2.51 ± 0.30	2.86 ± 0.00	2.69 ± 0.01	3.21 ± 0.31	3.64 ± 0.46	3.14 ± 0.10
**Aspartic acid**	3.45 ± 0.13	3.17 ± 0.35	3.73 ± 0.01	3.36 ± 0.14	4.05 ± 0.32	4.59 ± 0.51	4.06 ± 0.18
**Cysteine**	0.39 ± 0.01	0.38 ± 0.00	0.41 ± 0.00	0.42 ± 0.05	0.50 ± 0.02	0.49 ± 0.03	0.46 ± 0.01
**Glutamic acid**	6.02 ± 0.17	5.37 ± 0.45	6.16 ± 0.05	6.14 ± 0.21	7.38 ± 0.42	7.82 ± 0.86	7.06 ± 0.36
**Glycine**	2.83 ± 0.07	2.57 ± 0.23	2.74 ± 0.07	2.53 ± 0.15	2.88 ± 0.01	3.11 ± 0.37	2.82 ± 0.17
**Proline**	3.03 ± 0.09	2.73 ± 0.23	3.16 ± 0.05	3.03 ± 0.05	3.62 ± 0.18	3.87 ± 0.47	3.56 ± 0.16
**Serine**	2.68 ± 0.10	2.50 ± 0.17	2.84 ± 0.04	2.76 ± 0.09	3.46 ± 0.02	3.33 ± 0.44	3.21 ± 0.16
**Tyrosine**	1.30 ± 0.04	1.41 ± 0.11	1.76 ± 0.08	1.25 ± 0.07	1.52 ± 0.01	1.54 ± 0.12	1.62 ± 0.07
** *Fatty acid composition* **							
**12:0**	-	3.92 ± 0.00	7.80 ± 0.01	0.39 ± 0.00	0.17 ± 0.00	2.05 ± 0.03	4.03 ± 0.02
**14:0**	2.11 ± 0.01	2.49 ± 0.00	3.00 ± 0.00	2.02 ± 0.00	1.85 ± 0.01	2.19 ± 0.02	2.45 ± 0.01
**16:0**	18.03 ± 0.15	16.92 ± 0.00	16.34 ± 0.02	16.93 ± 0.04	16.26 ± 0.01	16.96 ± 0.09	16.17 ± 0.06
**16:1n−9**	3.56 ± 0.02	3.34 ± 0.00	3.22 ± 0.01	3.37 ± 0.01	3.17 ± 0.00	3.32 ± 0.01	3.16 ± 0.01
**18:0**	5.54 ± 0.05	4.95 ± 0.00	4.41 ± 0.01	5.27 ± 0.00	5.13 ± 0.00	5.13 ± 0.02	4.73 ± 0.02
**18:1n−9**	35.05 ± 0.28	34.63 ± 0.01	33.58 ± 0.12	36.67 ± 0.05	38.00 ± 0.07	36.02 ± 0.00	35.24 ± 0.11
**18:2n−6**	13.52 ± 0.08	14.43 ± 0.01	15.35 ± 0.03	14.40 ± 0.01	15.52 ± 0.02	14.82 ± 0.01	15.34 ± 0.03
**18:3n−3**	2.11 ± 0.01	2.21 ± 0.00	2.23 ± 0.01	2.31 ± 0.00	2.44 ± 0.00	2.27 ± 0.01	2.35 ± 0.01
**20:4n−6**	0.85 ± 0.00	0.68 ± 0.00	0.54 ± 0.00	0.74 ± 0.00	0.65 ± 0.00	0.70 ± 0.00	0.61 ± 0.00
**20:5n−3**	2.34 ± 0.00	1.87 ± 0.00	1.52 ± 0.00	2.16 ± 0.00	1.96 ± 0.02	1.92 ± 0.04	1.81 ± 0.03
**22:6n−3**	6.00 ± 0.06	4.49 ± 0.01	3.32 ± 0.01	5.16 ± 0.01	4.40 ± 0.05	4.61 ± 0.14	4.09 ± 0.11
**TOTAL SFA**	27.19 ± 0.22	29.61 ± 0.01	32.87 ± 0.04	26.01 ± 0.05	24.90 ± 0.02	27.65 ± 0.16	28.60 ± 0.13
**TOTAL MUFA**	41.36 ± 0.26	40.42 ± 0.01	38.78 ± 0.13	42.79 ± 0.06	43.88 ± 0.07	41.88 ± 0.01	40.88 ± 0.10
**TOTAL PUFA**	26.96 ± 0.02	25.60 ± 0.00	24.48 ± 0.06	26.92 ± 0.01	27.12 ± 0.08	26.30 ± 0.18	26.17 ± 0.13
**Lc-PUFA**	9.20 ± 0.02	7.03 ± 0.02	8.05 ± 0.01	7.23 ± 0.01	5.37 ± 0.02	7.01 ± 0.05	6.51 ± 0.05

Total SFA include 6:0, 8:0, 10:0, 12:0, 14:0, 15:0, 16:0, 17:0, 18:0, 22:0; Total MUFA include 16:1n−9, 17:1n−7, 18:1n−9, 20:1n−9; Total PUFA include 18:2n−6, 18:3n−6, 20:2n−6, 20:3n−6, 20:3n−3, 20:4n−6, 20:5n−3, 22:6n−3, Lc-PUFA: 20:4n−6 + 20:5n−3 + 22:6n−3.

**Table 4 animals-13-00968-t004:** Primers used in real-time quantitative PCR.

Target Gene	Forward Primer (5′–3′)	Reverse Primer (5′–3′)	Primer Efficiency	Reference
*fas*	TTGTTGCTCAAACCCAACACC	AGACTTGCAGGCTCCATCAG	1.95	MH253823
*fads2*	TCCTATATGCTGGAACTAATGTGG	AGGATGTAACCAACAGCATGG	1.95	[55]
*elovl2*	GCAGTACTCTGGGCATTTGTC	GGGACATTGGCGAAAAAGTA	1.91	[55]
*elovl5*	ACTCACAGTGGAGGAGAGC	GGAATGGTGGTAAACGTGCA	1.93	[55]
*elf1α*	CCTTCAACGCTCAGGTCATC	TGTGGGCAGTGTGGCAATC	1.94	[56]

*fas* fatty acid synthase; *fads2*, fatty acyl desaturase 2; *elovl2* elongase of very long-chain fatty acid 2; *elovl5* elongase of very long-chain fatty acid 5; *elf1α,* elongation factor-1 alpha.

**Table 5 animals-13-00968-t005:** Production performance (n = 60/diet group) and protein utilization efficiency (n = 9/diet group) of African catfish (mean ± SD).

Indicator ^1^	C	B50	B100	M50	M100	BM50	BM100	*Compar. B* ^2^		*Compar. M* ^2^		*Compar. BM* ^2^	
								*p_ANOVA_* ^3^	*p_linear_* ^4^	*p_ANOVA_* ^3^	*p_linear_* ^4^	*p_ANOVA_* ^3^	*p_linear_* ^4^
**FBW (g)**	**267.4 ± 59.3 ^e^**	257.6 ± 48.1	258.6 ± 47.0	**260.9 ± 41.5 ^de^**	**251.6 ± 49.5 ^d^**	263.8 ± 42.1	251.3 ± 39.7	0.422	0.291	0.087	**0.037**	0.215	0.105
**SGR (%day^−1^)**	**0.69 ± 0.06 ^eh^**	0.62 ± 0.07	0.60 ± 0.08	**0.61 ± 0.05 ^de^**	**0.54 ± 0.05 ^d^**	**0.65 ± 0.04 ^gh^**	**0.53 ± 0.02 ^g^**	0.359	0.187	**0.037**	**0.013**	0.069	**0.029**
**PER**	**0.98 ± 0.09 ^e^**	0.8 ± 0.11	0.84 ± 0.11	**0.86 ± 0.06 ^de^**	**0.74 ± 0.07 ^d^**	0.91 ± 0.17	0.77 ± 0.01	0.264	0.137	**0.021**	**0.007**	0.130	0.054
**PPV (%)**	**16.38 ± 0.49 ^e^**	15.66 ± 0.11	12.91 ± 0.11	**15.36 ± 0.06 ^e^**	**11.86 ± 0.07 ^d^**	15.67 ± 0.17	14.69 ± 0.01	0.237	0.118	**0.005**	**0.002**	0.723	0.441
**FCR**	**2.30 ± 0.23 ^eg^**	2.57 ± 0.30	2.68 ± 0.39	**2.59 ± 0.20 ^de^**	**2.97 ± 0.29 ^d^**	**2.47 ± 0.36 ^gh^**	**2.96 ± 0.13 ^h^**	0.379	0.191	**0.040**	**0.015**	**0.045**	**0.019**

^1^ FBW: final body weight, SGR: specific growth rate, PER: protein efficiency ratio, PPV: protein productive value, FCR: feed conversion ratio. ^2^ Three group comparisons were formed: Comparison B, between Control, B50, and B100; Comparison M, Control, M50, and M100; Comparison BM, Control, BM50, and BM100. ^3^
*p*-values of ANOVA in respective comparison group. Values in the same row that share same superscripts are not statistically different (*p* > 0.05) according to Tukey’s multiple comparison test. The letters ^d, e^ denote significant differences among treatments in Comparison M; letters ^g, h^ denote significant differences among treatments in Comparison BM; ^4^
*p*-values of linear components of the polynomial contrast analysis between each performance indicator and fish meal replacement level (0%–50%–100%) with insect-based ingredient. Second-order (quadratic) components were nowhere significant; therefore, these values are not listed.

**Table 6 animals-13-00968-t006:** Biometric index of African catfish (n = 18/diet group), (mean ± SD).

Indicator ^1^	C	B50	B100	M50	M100	BM50	BM100	*Compar. B ^2^*		*Compar. M ^2^*		*Compar. BM ^2^*	
								*p_ANOVA_ ^3^*	*p_linear_ ^4^*	*p_ANOVA_ ^3^*	*p_linear_ ^4^*	*p_ANOVA_ ^3^*	*p_linear_ ^4^*
**CF (g cm^−3^)**	0.83 ± 0.10	0.76 ± 0.07	0.78 ± 0.12	0.76 ± 0.06	0.82 ± 0.10	0.78 ± 0.05	0.78 ± 0.07	0.147	0.160	0.075	0.850	0.115	0.054
**HSI (%)**	1.57 ± 0.45	1.63 ± 0.40	1.70 ± 0.39	1.77 ± 0.37	1.73 ± 0.54	1.64 ± 0.42	1.73 ± 0.49	0.621	0.332	0.369	0.281	0.557	0.282
**TL (cm)**	32.08 ± 0.96	33.22 ± 2.24	32.91 ± 3.11	33.53 ± 1.24	31.50 ± 2.03	32.58 ± 1.53	32.61 ± 1.56	0.516	0.285	0.123	0.243	0.438	0.255

^1^ TL: total length, HSI: hepato-somatic index, CF: condition factor. ^2^ Three group comparisons were formed: Comparison B, between Control, B50, and B100; Comparison M, Control, M50, and M100; Comparison BM, Control, BM50, and BM100. ^3^
*p*-values of ANOVA in respective comparison group. ^4^
*p*-values of linear components of the polynomial contrast analysis between each performance indicator and fish meal replacement level (0%–50%–100%). Second-order (quadratic) components were nowhere significant; therefore, these values are not listed.

**Table 7 animals-13-00968-t007:** Selected blood plasma biochemistry parameters of African catfish fed diets with different inclusion level of M and B (n = 9/diet groups).

Indicator ^1^	C	B50	B100	M50	M100	BM50	BM100	*Compar. B ^2^*		*Compar. M ^2^*		*Compar. BM ^2^*	
								*p_ANOVA_ ^3^*	*p_linear_ ^4^*	*p_ANOVA_ ^3^*	*p_linear_ ^4^*	*p_ANOVA_ ^3^*	*p_linear_ ^4^*
**ALP** (U L^−1^)	45.8 ± 7.3	42.9 ± 3.6	44.5 ± 10.5	46.5 ± 5.1	45.7 ± 8.1	46.3 ± 6.4	46.0 ± 5.6	0.720	0.730	0.967	0.981	0.988	0.943
**ALT** (U L^−1^)	29.0 ± 18.5	30.5 ± 9.0	22.7 ± 7.8	38.6 ± 23.2	27.4 ± 11.8	34.0 ± 12.6	24.0 ± 11.7	0.927	0.816	0.365	0.663	0.547	0.850
**CA** (mg dL^−1^)	12.1 ± 2.01	14.1 ± 1.37	12.5 ± 2.1	12.44 ± 1.46	13.37 ± 1.30	12.6 ± 1.68	13.02 ± 1.89	0.078	0.650	0.368	0.172	0.590	0.310
**CHOL** (mg dL^−1^)	**115.2 ± 15.6 ^dg^**	125.4 ± 9.1	125.5 ± 13.7	**133.8 ± 11.3 ^de^**	**146.1 ± 18.9 ^e^**	**137.8 ± 14.2 ^i^**	**136.9 ± 13.9 ^h^**	0.187	0.120	**0.001**	**0.000**	**0.012**	**0.006**
**CREA** (mg dL^−1^)	0.21 ± 0.11	0.22 ± 0.08	0.17 ± 0.05	0.15 ± 0.08	0.19 ± 0.06	0.20 ± 0.10	0.23 ± 0.05	0.324	0.247	0.301	0.776	0.739	0.596
**GLOB** (g dL^−1^)	**2.45 ± 0.21 ^g^**	2.60 ± 0.31	2.63 ± 0.45	2.53 ± 0.19	2.70 ± 0.35	**2.70 ± 0.14 ^gh^**	**2.81 ± 0.26 ^h^**	0.500	0.275	0.159	0.063	**0.007**	**0.002**
**GLU** (mg dL^−1^)	101.5 ± 31.2	76.9 ± 13.6	94.6 ± 20.3	82.1 ± 22.5	81.2 ± 12.2	82.7 ± 15.7	82.6 ± 18.4	0.085	0.541	0.125	0.066	0.175	0.100
**GGT** (U L^−1^)	<5	<5	<5	<5	<5	<5	<5	-	-	-	-	-	-
**TP** (g dL^−1^)	3.19 ± 0.56	3.06 ± 0.37	3.15 ± 0.58	3.30 ± 0.34	3.37 ± 0.65	3.20 ± 0.44	3.28 ± 0.83	0.849	0.882	0.692	0.398	0.930	0.731
**PHOS** (mg dL^−1^)	**7.11 ± 1.36 ^be^**	**5.83 ± 0.41 ^a^**	**6.65 ± 0.76 ^ab^**	**5.91 ± 0.45 ^d^**	**5.85 ± 0.70 ^d^**	6.31 ± 0.91	6.41 ± 0.70	**0.027**	0.323	**0.013**	**0.007**	0.249	0.168
**IG** (g dL^−1^)	1.31 ± 0.72	0.80 ± 0.46	1.34 ± 0.92	0.83 ± 0.60	1.37 ± 0.96	1.56 ± 0.67	1.47 ± 0.89	0.162	0.931	0.193	0.858	0.724	0.619
**AMY** (U L^−1^)	21.4 ± 7.7	16.1 ± 3.3	24.1 ± 9.1	22.1 ± 10.4	16.0 ± 3.4	19.7 ± 3.7	18.1 ± 5.3	0.122	0.442	0.262	0.187	0.525	0.262

^1^ ALP: alkaline phosphatase, ALT: alanine aminotransferase, CA: calcium, CHOL: total cholesterol, CREA: creatinine, GLOB: globulin, GLU: glucose, GGT: gamma-glutamyl transpeptidase, TP: total protein; PHOS: phosphate, IG: immunoglobulin, AMY: amylase. Mean ± SD (n = 12 fish/group). ^2^ Three group comparisons were formed: Comparison B, between Control, B50, and B100; Comparison M, Control, M50, and M100; Comparison BM, Control, BM50, and BM100. ^3^
*p*-values of ANOVA in respective comparison group. Values in the same row that share same superscripts are not statistically different (*p* > 0.05) according to Tukey’s multiple comparison test. The letters ^a, b^ denote significant differences among treatments in Comparison B; The letters ^d, e^ denote significant differences among treatments in Comparison M; letters ^g, h, i^ denote significant differences among treatments in Comparison BM; ^4^
*p*-values of linear components of the polynomial contrast analysis between each performance indicator and FM inclusion level (0%–50%–100%). Second-order (quadratic) components were nowhere significant; therefore, these values are not listed.

**Table 8 animals-13-00968-t008:** Total lipid (% wet weight) and fatty acid composition (% of total fatty acids) of liver after feeding period using diets with different inclusion levels of M and B (n = 6/diet group).

Indicator ^1^	C	B50	B100	M50	M100	BM50	BM100	*Compar. B ^2^*		*Compar. M ^2^*		*Compar. BM ^2^*	
								*p_ANOVA_ ^3^*	*p_linear_ ^4^*	*p_ANOVA_ ^3^*	*p_linear_ ^4^*	*p_ANOVA_ ^3^*	*p_linear_ ^4^*
**Total lipid**	6.44 ± 2.65	8.09 ± 3.75	7.31 ± 2.51	5.78 ± 2.05	9.12 ± 2.42	5.64 ± 3.25	7.87 ± 2.80	0.648	0.626	0.064	0.070	0.427	0.410
**12:0**	-	0.22 ± 0.04	0.41 ± 0.08	-	-	-	0.31 ± 0.03	-	-	-	-	-	-
**14:0**	**1.27 ± 0.26 ^a^**	1.51 ± 0.24 ^ab^	**1.70 ± 0.18 ^b^**	1.58 ± 0.11	1.42 ± 0.21	1.56 ± 0.29	1.53 ± 0.20	**0.018**	0.802	0.055	0.205	0.123	0.094
**16:0**	28.34 ± 3.12	29.90 ± 0.90	30.15 ± 2.05	28.31 ± 2.35	29.93 ± 1.58	29.38 ± 2.96	30.82 ± 1.33	0.336	0.563	0.440	0.276	0.283	0.120
**16:1n−9**	3.44 ± 0.53	3.44 ± 0.31	3.05 ± 0.17	3.82 ± 0.55	3.49 ± 0.50	3.46 ± 0.53	3.43 ± 0.55	0.137	0.297	0.418	0.864	0.994	0.966
**18:0**	7.71 ± 1.57	10.23 ± 1.95	9.67 ± 2.18	7.44 ± 0.73	7.36 ± 0.66	8.63 ± 1.15	8.31 ± 2.15	0.089	0.128	0.837	0.574	0.640	0.549
**18:1n−9**	34.93 ± 6.08	37.60 ± 2.46	35.62 ± 1.65	37.31 ± 0.56	37.68 ± 1.71	36.42 ± 4.48	38.36 ± 2.86	0.488	0.253	0.393	0.213	0.461	0.222
**18:2n−6**	4.52 ± 1.54	4.18 ± 1.67	4.89 ± 1.67	6.02 ± 0.98	5.46 ± 1.04	4.79 ± 1.48	4.60 ± 1.60	0.755	0.529	0.130	0.201	0.953	0.934
**18:3n−6**	0.38 ± 0.14	0.40 ± 0.23	0.41 ± 0.13	0.59 ± 0.10	0.58 ± 0.15	0.56 ± 0.19	0.47 ± 0.13	0.392	0.282	0.602	0381	0.183	0.839
**18:3n−3**	0.43 ± 0.07	0.29 ± 0.17	0.32 ± 0.16	0.47 ± 0.11	0.40 ± 0.14	0.35 ± 0.10	0.31 ± 0.12	0.392	0.282	0.602	0.381	0.183	0.839
**20:4n−6**	1.22 ± 0.43	1.01 ± 0.50	1.35 ± 0.37	1.12 ± 0.26	1.13 ± 0.43	0.74 ± 0.32	0.84 ± 0.41	0.426	0.229	0.863	0.750	0.115	0.165
**20:5n−3**	0.54 ± 0.26	0.45 ± 0.17	0.25 ± 0.01	0.61 ± 0.24	0.29 ± 0.01	0.39 ± 0.21	0.21 ± 0.02	0.330	0.154	0.109	0.146	0.327	0.955
**22:6n−3**	**3.02 ± 1.15 ^h^**	2.49 ± 0.94	2.64 ± 0.74	3.03 ± 0.95	2.14 ± 0.81	**1.75 ± 0.72 ^gh^**	**1.60 ± 0.58 ^g^**	0.625	0.488	0.226	0.141	**0.021**	**0.011**
**Total SFA**	39.24 ± 2.19	42.55 ± 1.94	42.59 ± 3.62	38.09 ± 1.98	39.72 ± 1.55	40.77 ± 2.05	41.66 ± 2.02	0.077	0.243	0.347	0.673	0.162	0.063
**Total MUFA**	40.29 ± 6.43	42.81 ± 2.58	40.65 ± 1.63	43.09 ± 0.82	43.23 ± 2.00	41.61 ± 5.09	43.80 ± 3.08	0.532	0.273	0.366	0.214	0.494	0.248
**Total PUFA**	11.68 ± 4.45	10.08 ± 3.90	11.55 ± 3.22	13.65 ± 2.51	11.60 ± 1.78	9.91 ± 3.07	9.37 ± 3.51	0.736	0.443	0.456	0.966	0.544	0.299
**n−6/n−3**	**2.15 ± 0.50 ^adg^**	**2.41 ± 0.42 ^ab^**	**2.99 ± 0.95 ^b^**	**2.60 ± 0.77 ^e^**	**3.25 ± 0.94 ^e^**	**3.33 ± 1.09 ^h^**	**3.37 ± 0.60 ^h^**	**0.047**	**0.795**	**0.005**	**0.003**	**0.001**	**0.001**
**Lc-PUFA**	**4.51 ± 1.80 ^beh^**	**3.79 ± 1.60 ^a^**	**4.08 ± 1.21 ^a^**	**4.65 ± 1.45 ^d^**	**3.28 ± 1.34 ^d^**	**2.63 ± 1.15 ^g^**	**2.37 ± 1.07 ^g^**	**0.001**	**0.020**	**0.001**	**0.000**	**0.000**	**0.000**

^1^ Total SFA include 6:0, 8:0, 10:0, 12:0, 14:0, 15:0, 16:0, 17:0, 18:0, 22:0; Total MUFA include 16:1n−9, 17:1n−7, 18:1n−9, 20:1n−9; Total PUFA include 18:2n−6, 18:3n−6, 20:2n−6, 20:3n−6, 20:3n−3, 20:4n−6, 20:5n−3, 22:6n−3, n−6/n−3: Total n−6 PUFA/Total n−3 PUFA; Lc-PUFA: 20:4n−6+ 20:5n−3+ 22:6n−3. ^2^ Three group comparisons were formed: Comparison B, between Control, B50, and B100; Comparison M, Control, M50, and M100; Comparison BM, Control, BM50, and BM100. ^3^
*p*-values of ANOVA in respective comparison group. Values in the same row that share same superscripts are not statistically different (*p* > 0.05) according to Tukey’s multiple comparison test. The letters ^a, b^ denote significant differences among treatments in Comparison B; letters ^d, e^ denote significant differences among treatments in Comparison M; letters ^g, h^ denote significant differences among treatments in Comparison BM; ^4^
*p*-values of linear components of the polynomial contrast analysis between each performance indicator and FM inclusion level (0%–50%–100%). Second-order (quadratic) components were nowhere significant; therefore, these values are not listed.

**Table 9 animals-13-00968-t009:** Pearson correlation coefficient between total lipid (fat) and HSI.

		Total Lipid	HSI
fat	Pearson Correlation	1	−0.308 *
	Sig. (2-tailed)		0.047
	N	42	42

* Correlation is significant at the 0.05 level (2-tailed).

## Data Availability

Data available on request due to restrictions, e.g., privacy or ethical reasons.

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
