# Peer review of "Physiological Response of Grower African Catfish to Dietary Black Soldier Fly and Mealworm Meal"

_animals, 2023, doi:10.3390/ani13060968_

Round 1
Reviewer 1 Report
As the authors stated, the main contribution of this manuscript deals with the size of fish (bigger than in other studies) and the inclusion of a blend od two insect meals. Surprisingly, there are no data on fillet composition, which from my point of view would provide an important scientific input.
With regard to the tittle, I feel that the term juvenile is not adequate for 200 g sized fish.
In the Introduction, the reference 12 (line 61) does not correspond to crude protein content in insects but to soybean meal.
In Materials and Methods:
- Line 110: please, check the word “scarifying”. I think it should be written “sacrificing”.
- Figure 1 is not necessary, as it is explained in the text.
- How did the authors collect the faeces from tanks? It is important to know it, because many information of the manuscript is related to food intake (FCR, PER, PPV).
- In the heading of Table 1, chemical composition must be deleted, as these data are included in Table 2.
In Results:
- Regarding to statistical analysis, polynomial contrast is recommended to know the tendency of results. So, in Results section, the sentences in lines 303-305 and 308-310 should be checked.
- In section 3.2, nothing is written about Ca.
- In section 3.3 (line 340), no significant differences were found between “fishes fed with the different diets” not between the “diets”.
- Although Table 3 includes the AA content of diets, nothing is said in results.
An important part of the Discussion section centers on the digestibility of insect meals. However, no measurements of this parameter were performed during the study. Therefore, it is not possible to carry out an adequate discussion. In fact, the text corresponding to lines 391-416 is a bibliographic review rather than a discussion in which the results are compared with those obtained by other researchers.
In lines 424-425, authors state that “mortality was related more to the handling stress than the feed”. However, this information is not in the Results section and authors should explain what happened with the fishes.
The references in line 394 [32] and line 396 [38] are exchanged, please rewrite.
In lines 452 to 457, authors compare their results with other obtained with defatted mealworm meal, but at the end of the sentence make mention to fully fatted meal. Please, rewrite this sentence. Also, it will be interesting to know if the insect meals used in this experiment were defatted or not.
The last paragraph in the Discussion is confusing and does not reflect the results of this experiment, where the digestibility of the diets has not been estimated.
Author Response
Please see in the attachment.

Reviewer 2 Report
The authors have presented a paper with a currently very important line of research in aquaculture, such as the reduction of fishmeal in fish feed. Insects are undoubtedly a future food source for aquaculture production. The work is well planned and is new in the kind of study, but some statements made and some clarifications in the material and methods should be answered to improve the quality of the work.
Simple summary
End the paragraph with . not with ,
Summary:
Los autores indicant lo siguiente:
The results revealed similar protein apparent digestibility coefficients compared to the control diet in diets containing different levels of B and M meals, but lower level was observed when the insect meals were used simultaneously in the diet.
In the work, no study is made, or at least it is not indicated as in the material and methods of how the protein apparent digestibility is evaluated. This analysis is usually done in feces, with an internal marker in the feed.
Introduccion
Review the citations, sometimes they are separated with a space and in others not, put as indicated by the instructions for authors.
Line 49, there is a space between the number and the ,
Material y metodos:
The heading of table 3 indicates this “SUM: Total SFA + Total MUFA + Total PUFA”, but this data does not appear in the table
Line 228. Why is it indicated that plasma or mucus was used? Then it is not differentiated in the analyzes nor is it commented on in the discussion.
The total number of animals in the study was 420 but most analyzes were done with relatively low n. You could justify or indicate the reason for that n.
Has the total protein content of the fish been analyzed to calculate PPV? I can't find the analysis or the results of protein digestion in the work, but if the discussion talks about protein digestibility...
Results:
Table 7: Different letters in the same rows in CHO, do not seem to be well placed. B100 and BM50 should be ab not ac?
Line 350. The correlation of Pearson has not been explained in the statistics.
Disusion:
The authors give a lot of information about a possible effect of chitin and palatability in some of the treatments that lead to reduced growth. But the reality is that their results indicate that this effect does not exist (there are no significant differences) therefore this paragraph should be oriented in another way (line 389-410).
Line 438. But in this study there has not been a problem of palatability, since all the fish have had the same consumption and not my growth?
Paragraph 440-448. if there are no significant differences between treatments in glucose levels, a higher or lower number value is not due to treatment but to chance. This paragraph should be reformulated.
Be careful when stating that having more fat in the diet can lead to an increase in cholesterol, because in this study the diet richest in fat (BM100) is not the one with the highest value of cholesterol. I believe that these differences are due more to chance, and that a larger number of animals per sample would have made the differences insignificant.
There is no dispute about the levels of fatty acids in the liver, and the effect of diet on them, and it is a part that the authors say is the most important since insect diets are deficient in certain fatty acids.
Line 484. The authors continually talk about digestibility, but an analysis of digestibility as such is not made, only calculations such as PPV are made, but total protein values etc. of the fish before and after the trial are not indicated.
Author Response
Please see in the attachment.

Reviewer 3 Report
Excellent job on this research effort. My suggestion in the future is to include the economic analysis regarding the alternative of fish meal. This is because a research finding, is not only based on a good result, but it must tally with the related industry demand. Then I can conclude the alternative to this fish meal is a success.

Author Response
Thank you for your suggestion and comments. I agree that economic evaluation of new type of diets is important for the feed industry. In this study we where focusing to the physiological response of fish. We hope that the price of insects products will be more economical available than today.
Round 2
Reviewer 1 Report
The authors have made changes to most of the suggestions made in the first revision. However, there are still some points that need to be clarified in the manuscript.
In Material and Methods, it is not described how faeces and unused feed were collected. As it is stated in the text the feed was distributed by hand five times per day and thus depending on how often the faeces and food were removed, a part could disintegrate in the water, overestimating the amount of food eaten. Considering that an important part of the manuscript deals with food intake, an accurate description of the method must be described.
Data on survival rates are not included in the Results. Also, in the former version mortality was only mentioned in the Discussion and handling stress was considered a possible cause. Surprisingly, this information has disappeared in the manuscript. Is there an explanation for this?
Although fishes are under market size (final body weight ranged between 251-267 g), I feel that fillet composition would have been interesting to know the trend in composition changes derived from the use of diets that include insect meal. Also, I do not found the relevance to provide the AA profile of diets in Results and not to give any explanation about it.
Author Response
Please see in the attachment.
